# Prospective Assessment of Serum Lipid Alterations in Chronic Hepatitis C Patients Treated with Direct Acting Antivirals: Insights Six Months Post Sustained Virological Response

**DOI:** 10.3390/medicina60081295

**Published:** 2024-08-10

**Authors:** Oana Koppandi, Dana Iovănescu, Bogdan Miuțescu, Alexandru Cătălin Motofelea, Oana Maria Jigău, Andreea Iulia Papoi, Călin Burciu, Eyad Gadour, Deiana Vuletici, Eftimie Miuțescu

**Affiliations:** 1Department of Gastroenterology, Faculty of Medicine, Western University “Vasile Goldiș” of Arad, Revoluţiei Boulevard 94, 310025 Arad, Romania; koppandi.oana@uvvg.ro (O.K.); oana_st17@yahoo.co.uk (O.M.J.); papoi.andreea@uvvg.ro (A.I.P.); calin.burciu@umft.ro (C.B.); miutescu.eftimie@uvvg.ro (E.M.); 2Multidisciplinary Doctoral School, Western University “Vasile Goldiș” of Arad, 310025 Arad, Romania; 3Department of Gastroenterology and Hepatology, Faculty of Medicine, “Victor Babeș” University of Medicine and Pharmacy Timișoara, Eftimie Murgu Square 2, 300041 Timișoara, Romania; miutescu.bogdan@umft.ro (B.M.); deiana.vuletici@umft.ro (D.V.); 4Advanced Regional Research Center in Gastroenterology and Hepatology, “Victor Babeș” University of Medicine and Pharmacy Timișoara, 300041 Timișoara, Romania; 5Department of Internal Medicine, Faculty of Medicine, “Victor Babeș” University of Medicine and Pharmacy, Eftimie Murgu Square 2, 300041 Timișoara, Romania; alexandru.motofelea@umft.ro; 6Department of Gastroenterology and Hepatology, King Abdulaziz Hospital-National Guard, Ahsa 31982, Saudi Arabia; eyadgadour@doctors.org.uk; 7Department of Internal Medicine, Faculty of Medicine, Zamzam University College, Khartoum 11113, Sudan

**Keywords:** chronic hepatitis C, lipids, liver fibrosis, direct-acting antiviral agents

## Abstract

*Background and Objectives:* Chronic hepatitis C virus (HCV) infection is intricately linked with dysregulation of lipid metabolism. In particular, cholesterol plays a crucial role in HCV replication. Direct-acting antiviral agents (DAAs) therapy has revolutionized the hepatitis C treatment landscape, achieving high rates of sustained virological response (SVR). However, viral clearance comes with some alterations in lipid-related markers. This prospective study aimed to evaluate the impact of HCV clearance on lipid homeostasis and non-invasive liver fibrosis markers in hepatitis C patients treated with DAAs. *Material and Methods*: Fifty-two patients with varying degrees of fibrosis treated with DAAs therapy were evaluated at baseline and 24 weeks post-SVR. Lipid profiles and non-invasive liver fibrosis markers were assessed. *Results*: Our findings revealed an increase in total cholesterol, triglyceride, and LDLc (low-density lipoprotein cholesterol) levels at 24 weeks post-SVR, alongside an improvement in serum liver enzymes. Although improvements in liver stiffness were observed in non-invasive tests, there was an increase in lipid-related markers post-SVR. *Conclusions:* This suggests a potential increased cardiovascular risk despite improvements in liver function and fibrosis, highlighting the necessity for statin therapy in some cases and extended follow-ups for these patients. These findings underscore the importance of closely monitoring lipid profiles in chronic hepatitis C patients post-SVR, as well as the potential need for statin therapy to mitigate cardiovascular risk. Additionally, extended follow-up is essential to assess long-term outcomes and ensure the optimal management of these patients.

## 1. Introduction

Chronic HCV infection is one of the leading causes of liver-related morbidities and has been a significant public health concern, affecting millions of individuals worldwide. The advent of DAAs has shifted the treatment landscape with their promising high rates of SVR, offering bright prospects for the management of this disease. 

The World Health Assembly adopted the first ‘Global Health Sector Strategy on Viral Hepatitis’, aimed at eliminating viral hepatitis as a public health threat [1]. The number of newly diagnosed hepatitis C cases reported from countries across Europe remains at a high level, with a considerable variation between country-specific rates. An estimated 3.9 million individuals are chronically infected with HCV in EU/EEA countries, with national estimates of anti-HCV prevalence in the general population ranging from 0.1% to 5.9% [2].

In Central and Eastern Europe, the prevalence of anti-HCV antibodies varies between 0.27% and 3.5%, with the number of people infected with HCV in the general population being about 1.16 million [3,4]. Romania is considered to be the country with the highest prevalence of HCV in Europe, with various reported figures in recent decades (5.9% in 1990, 3.23% in 2010, and recently estimated at 2.7%, corresponding to 550,000 patients with viral loads) [5,6]. In accordance with the national protocol, genotype determination is not required for HCV treatment in Romania due to the high prevalence of genotype 1b (99.6%). This decision is supported by epidemiological data, which indicate that genotype 1b is predominant in the Romanian population [7].

Until recently, anti-HCV therapy was limited to interferon (IFN)-based regimens, which are associated with severe side effects and unsatisfactory cure rates [8]. The introduction of direct-acting antivirals brought a revolutionary change to anti-HCV therapy. Now, HCV can be eliminated from the infected host within 8–12 weeks of treatment without noticeable side effects, achieving SVR in >95% of cases [9]. 

HCV eradication with Peg-INF/Ribavirin was also associated over time with lipid metabolism alterations and lifestyle changes [10]. In the new era of DAAs therapy, the behavior of lipoproteins during the treatment and after achieving SVR is debatable. 

Clinical evidence indicates that HCV infection is not only intimately linked to the metabolism of lipids within the hepatocytes that HCV infects, but it dysregulates circulating lipoprotein metabolism as well. The liver is the central organ of lipid homeostasis for the entire body, producing and uptaking lipoproteins. Several studies have indicated a link between the successful outcome of the antiviral treatment and the observed lipid metabolism parameters of the patient [11].

It has been reported that alterations in lipid metabolism, such as an increase in LDL-cholesterol and a decrease in triglyceride levels, are observed after DAAs therapy [12]. A study published in 2020 on 394 patients showed increased serum LDLc levels after DAAs treatment in patients with chronic HCV infection, including those with genotype 6 [13]. A systematic review and meta-analysis that included 14 studies demonstrated that DAAs can induce lipid changes in patients with chronic hepatitis C, which may persist after treatment completion. These include a significant increase in LDLc, HDLc, and total cholesterol but no changes in triglycerides [14].

Since the increase in serum lipids has been reported at different magnitudes in the literature, the aim of the present study was to prospectively evaluate the impact of HCV clearance following direct-acting antiviral regimens on lipid homeostasis and liver fibrosis in our patients.

## 2. Materials and Methods

### 2.1. Patients 

In our prospective observational study, we enrolled 52 consecutive patients with chronic hepatitis C and compensated cirrhosis, regardless of prior treatment experience, who were admitted for treatment with DAA regimens approved in Romania (Ledipasvir/Sofosbuvir, Ombitasvir/Paritaprevir/Ritonavir/Dasabuvir, Glecaprevir/Pibrentasvir, and Sofosbuvir/Velpatasvir) at Arad County Emergency Clinical Hospital, Arad, Romania, between July 2022 and October 2023. The exclusion criteria were as follows: failure to follow up, decompensated cirrhosis, failure to achieve SVR, alcohol consumption up to 20 g/day for women and >30 g/day for men, coinfection of human immunodeficiency virus, and hepatocellular carcinoma. 

### 2.2. Demographic Data and Laboratory Tests

Demographic data, including age at inclusion, gender, ethnicity, and previous history of antiviral treatment, were recorded at baseline. At baseline and at follow-up visit at 24 weeks after SVR (SVR24), biochemical tests were performed, including complete blood count (CBC), aspartate 2-oxoglutarate aminotransferase (AST), alanine 2-oxoglutarate aminotransferase (ALT), albumin (Alb), total bilirubin (TB), direct bilirubin (DB), triglycerides (TG), total cholesterol (TC), low-density lipoprotein cholesterol, high-density lipoprotein cholesterol (HDLc), alkaline phosphatase (ALP), and gamma-glutamyl transferase (GGT). Fibrosis 4 (FIB-4) index was calculated according to the following equation: [age (years) × AST (U/L)]/[PLT (104/μL) × 10 × ALT (U/L)1/2]. Aspartate-aminotransferase-to-platelet ratio index (APRI) was assessed. Body mass index (BMI) was also assessed at baseline and SVR24. Fibrosis severity at baseline was assessed by Fibromax^®^ and by non-invasive liver tests (NITs) at baseline and at 24 weeks.

The patient’s history of dyslipidemia was ascertained, and the drug history regarding the prescription of lipid-lowering agents before, during, and after DAAs therapy was also recorded. There was no specific intervention regarding diet or lifestyle modification of the cohort throughout the treatment and follow-up period.

### 2.3. Statistical Analysis

Continuous variables were assessed for normality using the Shapiro–Wilk test. Normally distributed data were presented as means ± standard deviations (SDs), while non-normally distributed data were summarized using medians with interquartile ranges (IQRs; 25th to 75th percentiles). Categorical variables were described as counts and percentages. Differences between groups for normally distributed continuous variables were assessed using Welch’s *t*-test for comparisons between two groups and one-way ANOVA for multiple groups, incorporating post hoc tests (e.g., Tukey’s HSD) to pinpoint specific group differences. For non-parametric continuous data, the Mann–Whitney U test and Kruskal–Wallis test were applied for two and multiple groups, respectively, with subsequent Dunn’s post hoc analyses as necessary. Categorical data comparisons were conducted using Chi-square tests or Fisher’s exact test when expected cell counts were below five. Sample size calculations were conducted a priori to achieve a confidence level of 95% and a statistical power of 80% based on anticipated effect sizes and variance estimates derived from preliminary data. With 52 subjects, the power of our study is approximately 91.1%, which is above the conventional threshold of 80%. This indicates that our sample size is more than adequate to detect a clinically meaningful effect size of 0.5 with a high degree of confidence. All statistical analyses were performed in R (version 3.6.3), leveraging the capabilities of several comprehensive packages within the Tidyverse for data manipulation and visualization, Finalfit for regression analyses, and other specialized packages (MCGV, Stringdist, Janitor, Hmisc) for various data processing needs.

## 3. Results

### 3.1. Patient Characteristics

The demographic characteristics of the study cohort, including gender distribution, age profile, and BMI levels, provide crucial insights into potential factors influencing disease dynamics and treatment outcomes (Table 1).

The study involved a varied group of 52 patients, with a significant majority being female, indicating potential gender-specific aspects in the condition’s prevalence or treatment response. The mean age was 62.4 years, suggesting a predominantly elderly population, potentially impacting the study’s findings related to disease progression and treatment efficacy due to age-related factors. The average BMI stood at 26.39 kg/m^2^, indicating that the majority of the cohort was in the overweight category, a significant concern considering the association between excess weight and various comorbid conditions. BMI analysis further delineated the nutritional and metabolic status of the patients, with the majority of the patients being of normal weight but a substantial percentage falling into obesity class I. 

Transitioning from the demographic overview, the study also delved into the diverse treatment modalities employed. Patients received different treatment types: Ledipasvir/Sofosbuvir therapy was the most common, followed by Glecaprevir/Pibrentasvir therapy. This diversity in treatment regimens highlights the tailored approach to managing the condition, taking into account individual patient needs and disease characteristics. Moreover, alongside treatment types, the study considered treatment duration and the background of the study population. A considerable proportion of the cohort underwent an 8 week treatment regime, aligning with standard practices for the condition being treated.

Shifting from treatment details, the study then assessed the severity of liver fibrosis among treatment-naive patients. The study population was predominantly treatment-naive, suggesting the findings may primarily reflect the initial efficacy and impact of therapeutic interventions. Fibrosis assessment revealed a significant distribution across various stages, with a noteworthy percentage of patients experiencing advanced fibrosis (F4), underscoring the severity of the condition within the cohort.

The study also investigated other factors, such as residence. More than half of the study population resided in urban areas, possibly reflecting environmental or lifestyle factors pertinent to the condition’s prevalence or management. 

The most common comorbidities were essential hypertension and type 2 diabetes, highlighting the complex clinical profile of the patients and the need for integrated care strategies (Figure 1). 

Liver health, as measured by Fibroscan and Fibromax, indicated widespread liver disease within the cohort, with 56.8% of patients diagnosed with liver steatosis and a significant number presenting with varying degrees of fibrosis, emphasizing the critical need for effective therapeutic interventions targeting liver health.

### 3.2. Changes in Lipid-Related and Fibrosis Markers

When evaluating the effects of DAAs intervention at 24 weeks after SVR on various biochemical parameters (Table 2), we observed differential results in various metabolic and liver function markers. Notably, there was no significant change in hemoglobin (Hb) levels post-intervention: these levels changed slightly, indicating a minimal impact of the SVR24 intervention points to this marker. The study found changes in liver enzymes; AST and ALT levels showed a considerable reduction, highlighting the DAAs intervention’s effectiveness in improving liver enzyme activities. Furthermore, GGT levels decreased significantly, indicating a return to normal liver function after DAAs therapy. 

Moreover, while direct bilirubin and total bilirubin levels experienced significant reductions (*p* < 0.001 for both), indicating positive alterations in bilirubin metabolism, the albumin levels showed a non-significant increment, implying stable nutritional and liver statuses. In addition, alkaline phosphatase levels demonstrated a significant decrease (*p* = 0.004), reinforcing the DAA intervention’s beneficial impact on liver function. 

Lipid profile assessments revealed an escalation in total cholesterol levels after the intervention, alongside increases in triglycerides and low-density lipoprotein cholesterol. Despite these positive changes, differences in lipid profiles require careful monitoring and possibly corrective measures to mitigate increased cardiovascular risk.

A paired *t*-test was performed and revealed significant improvements in both APRI and FIB-4 scores after treatment. Specifically, the APRI score improved from baseline to 24 weeks, reflecting a statistically significant mean difference of 0.69 (SE = 0.122, *p* < 0.001). Similarly, the FIB-4 score showed a reduction after 24 weeks, with a significant mean difference of 1.046 (SE = 0.25, *p* < 0.001). These findings suggest that the treatment was effective in improving the conditions as measured by the APRI and FIB-4 scores in our study (Table 3). These improvements can be partially attributed to the significant reduction in transaminase levels, which is a key component of both scoring systems.

During the follow-up period, we also observed changes in BMI that correlated with lipid changes. Specifically, the mean BMI of patients slightly increased from 26.4 kg/m^2^ at baseline to 27 kg/m^2^ after treatment.

Among the 52 patients in our study, there was one major cardiovascular event after treatment, representing 1.9% of the cohort. Additionally, seven patients (13.5%) required statin therapy post-SVR to manage elevated lipid levels. These findings provide insight into the cardiovascular health and therapeutic needs of our patient population following the HCV cure.

Regarding treatment regimens, we observed some differences within the groups. We noticed an increase in total cholesterol levels from baseline to six months after treatment in SOF-based therapies, as well in the GLE/PIB group, and a decrease in values in the OBV/PTV/r + DSV group. Triglycerides showed a slight overall increase in all groups but with no clinical significance. On the other hand, HDLc levels increased after treatment in all groups except the OBV/PTV/r + DSV group, where there was a significant decrease in values. LDLc levels also showed differences between treatment groups at SVR 24, with values increasing above the normal range in SOF-based therapies groups as well as in the GLE/PIB group, and a positive alteration with normalization of the values in the OBV/PTV/r + DSV group (Table 4). 

## 4. Discussion

Directing our attention to the implications, the mechanism that generates the changes in lipid levels after viral clearance by DAA drugs is still unclear, but there are two possible explanations. One is the restoration of cholesterol synthesis that accompanies improved liver function after clearance of HCV infection, and the other is related to reduced liver inflammation [15].

Accordingly, the rapid increase in lipid levels after HCV clearance likely reflects the reversal of lipid metabolism perturbation by inhibiting HCV replication [16]. A study conducted by Corey et al. provides evidence that HCV infection can lead to significant alterations in lipid metabolism due to the virus’s interaction with lipid pathways. Following successful treatment and viral eradication, these lipid metabolism pathways can normalize, which may explain the lipid changes observed post-cure in our study [17].

Lipid homeostasis and the metabolic effects of interferon-free DAAs treatment during therapy and after SVR have recently been studied extensively in the literature. Villani et al. showed in a systematic review and meta-analysis, that included 14 studies (N = 1537 patients), an increase in total cholesterol 4 weeks after starting therapy (+15.86 mg/dL; 95% CI + 9.68 to 22.05; *p* < 0.001) and 12 weeks after treatment completion (+17.05 mg/dL; 95% CI + 11.24 to 22.85; *p* < 0.001). The LDL trend was similar to the total cholesterol change in the overall analysis. Triglycerides did not show significant changes during the treatment or after the treatment was completed. The study concluded that DAAs induce mild lipid changes in chronic hepatitis C patients, which may persist after treatment completion. The same study results showed that Sofosbuvir/Ledipasvir use was associated with a higher total cholesterol increase during treatment, which was comparable with our findings that showed higher total cholesterol levels and LDLc levels in the SOF-based therapy groups. These lipid changes observed at SVR 24 likely reflect the initial adjustments in lipid metabolism following the HCV eradication, although they may also be influenced by the limited DAA washout time. No significant increase in triglycerides was observed within any of the therapy groups [14].

The novelty of our study lies in a prospective comprehensive analysis of lipid profile dynamics, liver function, and several non-invasive liver fibrosis parameters, with a more extensive follow-up period after treatment completion, which can allow for the evaluation of the need for medical intervention on dyslipidemia. Our results were comparable, revealing a significant escalation in total cholesterol levels alongside increases in triglycerides and LDLc at SVR 24. The results are also consistent with Hashimoto et al. [18] and Mauss et al. [19], who reported an increase in total cholesterol and LDLc, but not in HDLc levels, following SVR with DAAs therapy, emphasizing the role of the intrahepatic cholesterol biosynthetic pathway in the HCV replication cycle. To strengthen this hypothesis, a prospective observational cohort study carried out on 301 chronic HCV-infected patients in Egypt compared lipid parameters during DAAs treatment and concluded that there was a statistically significant difference between responder and non-responder groups regarding baseline cholesterol (*p* = 0.001), LDLc (*p* = 0.001), HDLc (*p* = 0.026) and triglyceride levels (*p* = 0.016). These lipid parameters increased significantly over the course of the therapy and continued to increase in responders after the treatment was discontinued, while they showed non-significant changes in non-responders [20].

Also, Hashimoto et al. [18] suggested that in addition to having an anti-HCV effect, DAAs might have pharmacological effects on serum lipid levels, especially on LDLc.

In our study, based on a well-characterized cohort of patients with chronic HCV infection, SVR 24 was found to be associated with a significant improvement in liver function tests, and fibrosis scores APRI and FIB-4 showed decreasing trends, reinforcing that the DAAs intervention has a beneficial impact on liver function and the progression of liver fibrosis. The observed improvements in APRI and FIB-4 scores post-treatment are likely influenced by the significant reduction in transaminase levels achieved through therapy. Since transaminase levels are integral components of both scoring systems, their reduction can markedly impact the scores, reflecting not only changes in liver fibrosis but also the biochemical response to treatment.

Many studies have focused on the improvement of non-invasive tests (NITs) of liver fibrosis and liver stiffness (LS) measured by transient elastography (TE) after SVR. Bachofner et al. have shown a rapid decrease in LS as measured by TE after treatment with DAAs. At the same time, fibrosis scores (APRI and FIB-4) improved significantly [21]. More recently, in a Spanish study, Pons et al. evaluated the evolution of LS in more than 500 patients with HCV-associated compensated Advanced Chronic Liver Disease (cACLD) who obtained SVR after DAAs therapy. The authors observed that after 1 year of follow-up, the mean LS after SVR decreased by nearly 30% from baseline. This change in LS cannot be explained only by a decrease in actual fibrosis since the time course is too short for significant remodeling. Further long-term longitudinal evaluations of the dynamics of LS changes should be performed to consolidate these results and to assess if they are associated with a better disease outcome [22].

Our study primarily focused on the biochemical and metabolic changes, particularly lipid alterations, following the HCV cure. Regarding cardiovascular outcomes, we observed one major cardiovascular event in the group after treatment. Additionally, 13.5% of the patients required statin therapy post-SVR to manage elevated lipid levels. These findings, although limited, highlight the need for ongoing cardiovascular risk assessment and management in patients achieving SVR. Future research should include long-term follow-ups to evaluate the broader impact of SVR on cardiovascular outcomes and the effectiveness of various therapeutic strategies in reducing cardiovascular risk.

Several limitations were associated with the present study. First, there was a limited number of patients. Secondly, there are other parameters that might alter the results, such as a change in body weight, dietary modifications, and exercise amount, which were not available. Beyond that, a possible confounder when analyzing and interpreting the results could be the heterogeneity of our study population, which consists of different stages of liver fibrosis.

## 5. Conclusions

In conclusion, our results suggest a dynamic change in serum lipid parameters, with a significant increase in total cholesterol levels, increases in triglycerides and LDLc, and the significant and continuous reduction in liver stiffness by non-invasive fibrosis measurements in chronic hepatitis C after DAA-induced SVR at 24 weeks. Therefore, further studies are required to investigate whether these observations are maintained over a more extended follow-up period.

## Figures and Tables

**Figure 1 medicina-60-01295-f001:**
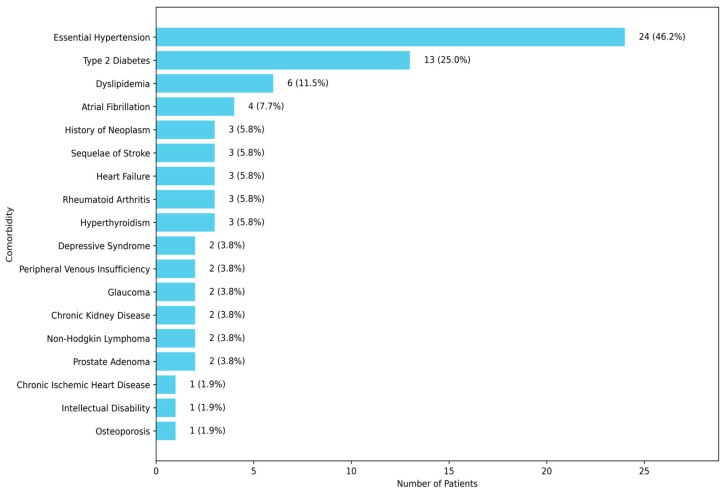
Prevalence of comorbid conditions.

**Table 1 medicina-60-01295-t001:** Baseline characteristics of the study population.

Variables	Statistics
Sex	
Female (%) Male (%)	31 (59.6%) 21 (40.4%)
Therapy (%)	
Sofosbuvir/Velpatasvir	9 (17.3%)
Ledipasvir/Sofosbuvir	20 (38.5%)
Glecaprevir/Pibrentasvir	17 (32.7%)
Ombitasvir/Paritaprevir/Ritonavir/Dasabuvir	6 (11.5%)
Duration (weeks) = 8 (%)	28 (53.8%)
Naive/Experienced = N (%)	46 (88.5%)
Fibrosis (Fibromax) (%)	
F0	3 (5.8%)
F1	5 (9.6%)
F1–F2	11 (21.2%)
F2	4 (7.7%)
F3	11 (21.2%)
F4	18 (34.6%)
Age (mean (SD))	62.44 (10.07)
BMI (%)	
Mildly Underweight	2 (3.8%)
Normal Weight	21 (40.4%)
Overweight	13 (25%)
Obesity Class I	15 (28.8%)
Obesity Class II	1 (1.9%)
Environment = U (%)	33 (63.5%)
Fibroscan (mean (SD))	8.36 (3.48)

**Table 2 medicina-60-01295-t002:** Impact of DAAs intervention on liver function and lipid profiles.

Variable	Baseline Mean (SD)	SVR24 Mean (SD)	Statistic	Mean Difference	*p*-Value
Hb	14.165 (1.669)	14.312 (1.411)	−0.789	−0.1462	0.434
AST	61.333 (35.933)	23.043 (7.223)	8.195	38.2892	<0.001
ALT	69.941 (47.800)	18.194 (7.736)	8.287	51.7463	<0.001
DB	0.363 (0.235)	0.230 (0.113)	4.546	0.1338	<0.001
TB	0.764 (0.584)	0.555 (0.313)	3.598	0.2090	<0.001
TC	162.245 (37.743)	182.855 (44.273)	−3.654	−20.6094	<0.001
TG	103.624 (41.389)	107.536 (41.339)	−0.688	−3.9119	0.494
HDLc	51.523 (14.147)	53.160 (18.248)	−0.914	−1.6363	0.365
LDLc	97.397 (32.565)	108.629 (34.127)	−2.937	−11.2315	0.005
ALB	4.268 (0.431)	4.184 (0.386)	0.995	0.0847	0.333
ALP	97.596 (49.910)	80.246 (26.571)	3.057	17.3500	0.004
GGT	82.897 (92.373)	33.300 (34.629)	4.148	49.5965	<0.001

**Table 3 medicina-60-01295-t003:** Comparative analysis of APRI and FIB-4 scores pre- and post-treatment.

Variable	Baseline Mean (SD)	SVR 24 Mean (SD)	Statistic	df	*p*-Value	Mean Difference	SE Difference
APRI	1.015 (0.966)	0.325 (0.233)	5.67	51	<0.001	0.69	0.122
FIB 4	3.075 (2.562)	2.029 (1.358)	4.18	51	<0.001	1.046	0.25

**Table 4 medicina-60-01295-t004:** Analysis of lipid profiles pre- and post-treatment in treatment regimen groups.

	SOF/VEL	LDV/SOF	GLE/PIB	OBV/PTV/r + DSV	Total	*p* Value
TC BL						0.3611
Mean (SD)	147.5 (30.3)	163.1 (36.2)	159.5 (39.1)	178.3 (30.5)	161.8 (35.8)	
Range	102.9–189.0	102.6–236.0	68.0–234.0	114.5–213.0	68.0–236.0	
TG BL						0.6691
Mean (SD)	89.1 (19.7)	106.3 (47.8)	103.2 (39.4)	113.4 (49.1)	103.9 (42.2)	
Range	68.2–129.0	41.0–267.0	58.0–202.0	68.0–208.0	41.0–267.0	
HDLc BL						0.7671
Mean (SD)	54.0 (15.6)	50.3 (13.9)	53.2 (14.7)	48.6 (6.3)	51.5 (13.6)	
Range	29.8–80.0	26.7–80.3	30.5–83.2	41.5–60.0	26.7–83.2	
LDLc BL						0.6491
Mean (SD)	88.8 (35.5)	95.1 (30.4)	97.1 (34.9)	108.6 (31.4)	96.5 (32.3)	
Range	42.0–138.0	45.8–147.0	25.6–170.0	46.9–150.0	25.6–170.0	
TC SVR24						0.4271
Mean (SD)	185.7 (37.8)	189.6 (45.0)	183.1 (39.4)	155.2 (62.6)	182.9 (44.3)	
Range	145.0–244.9	121.5–266.3	114.0–263.6	67.0–233.7	67.0–266.3	
TG SVR24						0.8991
Mean (SD)	103.8 (44.2)	103.9 (43.0)	110.6 (40.7)	116.6 (42.0)	107.5 (41.3)	
Range	61.7–208.4	57.3–229.7	61.3–200.1	83.6–200.0	57.3–229.7	
HDLc SVR24						0.1611
Mean (SD)	58.0 (20.2)	54.6 (18.5)	54.3 (16.4)	37.7 (15.5)	53.2 (18.2)	
Range	30.1–99.0	24.7–85.3	29.8–84.7	9.3–55.0	9.3–99.0	
LDLc SVR24						0.4641
Mean (SD)	104.2 (21.3)	117.3 (33.1)	106.0 (33.3)	94.0 (53.2)	108.6 (34.1)	
Range	71.6–140.4	42.5–170.2	49.4–168.6	18.0–156.0	18.0–170.2	

Abbreviations: SOF/VEL, Sofosbuvir/Velpatasvir; LDV/SOF, Ledipasvir/Sofosbuvir; GLE/PIB, Glecaprevir/Pibrentasvir; OBV/PTV/r + DSV, Ombitasvir/Paritaprevir/Ritonavir/Dasabuvir; BL, baseline; sustained viral response at 6 months.

## Data Availability

The data presented in this study are available at the request of the corresponding author for privacy reasons.

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
