# Peer review of "Prospective Assessment of Serum Lipid Alterations in Chronic Hepatitis C Patients Treated with Direct Acting Antivirals: Insights Six Months Post Sustained Virological Response"

_medicina, 2024, doi:10.3390/medicina60081295_

Round 1

Reviewer 1 Report

Comments and Suggestions for Authors

Prospective Assessment of Serum Lipid Alterations in HCV Patients Treated with DAAs: Insights Six Months Post-SVR

MATERIAL AND METHODS

Patients

the study included a small number of patients (52) , 35% of them have an advanced liver fibrosis

Demographic data and laboratory tests

SVR 6 can be indicated as SVR 24 (i.e. 24 weeks after end of therapy)

RESULTS

Patient Characteristics

×         line 143-144 and table 1: indicate

×         sofosbuvir /velpatasvir

×         ledipasvir/sofosbuvir

×         glecaprevir/pibrentasvir

×         Ombitasvir / Paritaprevir / Ritonavir / Dasabuvir

Instead of

×         Sofosbuvirum / Velpatasvirum

×         Ledipasvirum / Sofosbuvirum

×         Glecaprevirum / Pibrentasvirum

×         Ombitasvirum / Paritaprevirum / Ritonavirum / Dasabuvirum

Discussion

×         Improvements in both APRI and 196 FIB-4 scores after treatment can be influenced by transaminase significant reduction with the therapy

×         Line 247 typing error “..the results are also are consistent with..”

×         More than 50% of enrolled patients were overweight/obese and almost 57% had liver steatosis. Add some data on any change in body weight or steatosis during follow up that can correlate with lipid changes.

×         Any correlation with genotype, if available, can be evaluated

×         No data on any therapeutical approach and impact on cardiovascular events after SVR

Comments on the Quality of English Language

intermediate

Author Response

Dear Reviewer,

Thank you very much for your comments and suggestions on our manuscript entitled "Prospective Assessment of Serum Lipid Alterations in HCV Patients Treated with DAAs: Insights Six Months Post-SVR". We appreciate all of your work and time dedicated on our manuscript.  We hope this version of the manuscript meets your approval. All changes have been highlighted in the manuscript. 

Comments 1: The study included a small number of patients (52), 35% of them have an advanced liver fibrosis.

Response 1: Thank you for the observation. We acknowledge the limitations regarding the small sample size and the proportion of patients with advanced liver fibrosis in our study. Our study included a relatively small number of patients (52), with 35% having advanced liver fibrosis. Given the monocentric characteristic of our study, the generalizability of our findings is limited. Additionally, the lack of a widespread screening program in our region contributes to fewer patients seeking medical attention early, resulting in a higher percentage of patients presenting with advanced fibrosis.

Comments 2: Demographic data and laboratory tests SVR 6 can be indicated as SVR 24 (i.e. 24 weeks after end of therapy).

Response 2:  Thank you for the observation. We understand the importance of clarity in reporting the follow-up period for demographic data and laboratory tests. To align with standard terminology and improve clarity, we have revised our manuscript to indicate SVR6 data as SVR24, representing 24 weeks after the end of therapy. We have made these changes throughout the manuscript to ensure consistency. Lines: 28, 30, 100, 109, 179, 183, 198, 200, 253, 262, 277, 315, Table 2, Table 3, Table 4

Comments 3: Results - Patient Characteristics, line 143-144 and table 1: indicate sofosbuvir /velpatasvir, ledipasvir/sofosbuvir, glecaprevir/pibrentasvir, Ombitasvir / Paritaprevir / Ritonavir / Dasabuvir Instead of Sofosbuvirum / Velpatasvirum, Ledipasvirum / Sofosbuvirum, Glecaprevirum / Pibrentasvirum, Ombitasvirum / Paritaprevirum / Ritonavirum / Dasabuvirum.

Response 3: Thank you for the observation. We have updated the manuscript to use the correct drug names as you suggested. Lines: 151, 152 and Table 1

Comments 4: Improvements in both APRI and FIB-4 scores after treatment can be influenced by transaminase significant reduction with the therapy.

Response 4: Thank you for your insightful comment. We agree that the significant reduction in transaminase levels with therapy can influence the improvements observed in both APRI and FIB-4 scores after treatment. This is an important consideration, and we have addressed it in the revised manuscript to provide a more comprehensive understanding of the factors contributing to the observed improvements in these scores. We have added the following discussion to the Results and Discussion sections to reflect this point: Results – “These improvements can be partially attributed to the significant reduction in transaminase levels, which is a key component of both scoring systems”; Discussion – “The observed improvements in APRI and FIB-4 scores post-treatment are likely influenced by the significant reduction in transaminase levels achieved through therapy. Since transaminase levels are integral components of both scoring systems, their reduction can markedly impact the scores, reflecting not only changes in liver fibrosis but also the biochemical response to treatment." Lines: 203, 204 and 280-284

Comments 5: Line 247 typing error “..the results are also are consistent with..”

Response 5: Thank you for pointing out the typing error. We have corrected the sentence on line 247 (original mauscript). Line: 262

Comments 6: More than 50% of enrolled patients were overweight/obese and almost 57% had liver steatosis. Add some data on any change in body weight or steatosis during follow up that can correlate with lipid changes.

Response 6: Thank you for your valuable comment. We appreciate the opportunity to provide additional data on changes in body weight during follow-up, which can help correlate with lipid changes. We have analyzed the follow-up data and included the relevant findings in the manuscript. While we have detailed data on BMI changes, specific quantification of liver steatosis post-treatment was not conducted. We included the additional data in the results section. Lines: 108, 205-207

Comments 7: Any correlation with genotype, if available, can be evaluated

Response 7: Thank you for your suggestion. Unfortunately, genotype data is not available for our study cohort. Our national protocol in Romania does not require genotype determination due to the high prevalence of genotype 1b and the use of pangenotypic regimens. As a result, we could not evaluate correlations between specific HCV genotypes and lipid changes. We have added a statement in the Introduction section of the manuscript to reflect this limitation. Lines: 56-59

Comments 8: No data on any therapeutical approach and impact on cardiovascular events after SVR.

Response 8: Thank you for highlighting this important aspect. We do have some relevant data regarding cardiovascular outcomes. Among the 52 patients in our study, there was one major cardiovascular event after treatment, representing 1.9% of the cohort. Additionally, 7 patients (13.5%) required statin therapy post-SVR to manage elevated lipid levels. Although these numbers are small, they provide some insight into the cardiovascular health and therapeutic needs of our patient population following HCV cure. We have added this information to both the Results and Discussion sections of the manuscript. Lines: 208-212 and 297-304

Reviewer 2 Report

Comments and Suggestions for Authors

Dear Colleague,

thank you very much for presenting your data.

This is a well-planned and nicel written paper

Regardless of its premises I have some recommendation which you may find below,

1. Please check once again the name of DAAs, ie Ledipasvir instead of ledipasvirum

2. Line 138 (and vice versa) includes comments. Please remove them, as the results section should only include the results without any comments. The whole discussion section is dedicated for your comments

3.  kindly provide the data on genotype distribution and classify the prescribed DAAs according to HCV genotype

4. Please kindly put together all the enzyme changes together without interrupting the data on albumin, etc.

5. could the lipid changes after cure may result from HCV pathogenesis? (PMID: 23698400)

6. your data focused on SVR6 changes on lipids. DAA washout time is limited so that you may consider revising your sentence line 237-38

Author Response

Dear Reviewer,

Thank you very much for your comments and suggestions on our manuscript entitled "Prospective Assessment of Serum Lipid Alterations in HCV Patients Treated with DAAs: Insights Six Months Post-SVR". We appreciate all of your work and time dedicated on our manuscript.  We hope this version of the manuscript meets your approval. All changes have been highlighted in the manuscript. 

Comments 1: Please check once again the name of DAAs, ie Ledipasvir instead of ledipasvirum.

Response 1: Thank you for the observation, we changed the name of the DAAs from Sofosbuvirum / Velpatasvirum, Ledipasvirum / Sofosbuvirum, Glecaprevirum / Pibrentasvirum, Ombitasvirum / Paritaprevirum / Ritonavirum / Dasabuvirum to Sofosbuvir / Velpatasvir, Ledipasvir / Sofosbuvir, Glecaprevir / Pibrentasvir, Ombitasvir / Paritaprevir / Ritonavir / Dasabuvir. Line: Table 1.

Comments 2:  Line 138 (and vice versa) includes comments. Please remove them, as the results section should only include the results without any comments. The whole discussion section is dedicated for your comments.

Response 2: Thank you for the observation, we removed the comment.  Line: 145

Comments 3: Kindly provide the data on genotype distribution and classify the prescribed DAAs according to HCV genotype.

Response 3: Thank you for your insightful comment. We appreciate your suggestion and would like to provide some context specific to the Romanian healthcare protocol for HCV treatment. Unfortunately, we can’t provide data about genotype distribution because it’s unavailable, as the national protocol for HCV treatment does not require genotype determination. This decision is based on epidemiological data indicating that a high percentage of HCV cases in Romania are of genotype 1b. Consequently, routine genotyping is not performed, and thus, we do not have genotype distribution data for our study population. According to PMID: 29253053, genotype 1b is indeed predominant in Romania, which supports our protocol. The prescription practices in Romania are largely based on the high prevalence of genotype 1b. Therefore, the treatment regimens are chosen with this genotype in mind. Additionally, to simplify the treatment protocol and ensure broad efficacy, pangenotypic regimens are predominantly used in the present.

We have included this explanation in the Introduction section of our manuscript to clarify our approach and the rationale behind it. We hope this addresses your concern and enhances the understanding of our study’s context. Line: 56-59

Comments 4: Please kindly put together all the enzyme changes together without interrupting the data on albumin, etc.

Response 4: Thank you for your suggestion, we rearranged the data in order to consolidate all enzyme changes into a single section, ensuring that the information on albumin and other related parameters is presented without interruption. Line: 187-191

Comments 5:  Could the lipid changes after cure may result from HCV pathogenesis? (PMID: 23698400)

Response 5: Thank you for your insightful comment. We appreciate the opportunity to discuss the potential relationship between lipid changes after HCV cure and HCV pathogenesis. The cited study (PMID: 23698400) provides evidence that HCV infection can lead to alterations in lipid metabolism. These changes are likely due to the virus's interaction with lipid pathways, contributing to the overall pathogenesis of the disease. We have included a discussion on this potential mechanism in the Introduction section of our manuscript to provide a comprehensive understanding of the observed lipid changes and their possible link to HCV pathogenesis.

Also, the study by Corey et al. (PMID: 19787818) provides evidence that HCV infection can lead to significant alterations in lipid metabolism due to the virus's interaction with lipid pathways. We have also included a discussion on this potential mechanism in the Discussion section of our manuscript.

Line: 68-73, 235-240

Comments 6: Your data focused on SVR6 changes on lipids. DAA washout time is limited so that you may consider revising your sentence line 237-38.

Response 6: Thank you for your valuable comment. We appreciate the opportunity to clarify the relationship between SVR6 changes in lipid profiles and the limited DAA washout time. We acknowledge that the observed lipid changes at SVR6 could be influenced by the limited washout period of Direct Acting Antivirals (DAAs). To address this, we have revised the sentences on lines 237-38 (original manuscpript) to reflect these considerations. Thank you again for your insightful feedback. Lines: 253-256

Reviewer 3 Report

Comments and Suggestions for Authors

First of all, I would like to congratulate the authors for the very interesting work they have done. It is a pity that they did not have a larger sample size. Nevertheless, it opens the door to further research on the subject. Secondly, I would like to make several comments in order to improve the manuscript:

  • The title of an article should not contain acronyms or abbreviations, even if they are widely known. I am referring to HCV, DDAs, post-SVR. It is more appropriate to write the full terms in the title. In the abstract, they do it correctly, the first time the term appears, the abbreviation is added.
  • Within the abstract, the term "direct-acting antivirals" is not found in the MeSH keywords. They should find a term that is in the thesaurus. This is important for indexing the work.
  • In the materials and methods section, this section should begin by defining the type of study design used (whether it is an observational, cross-sectional, pre-post design, etc.).
  • In this same section, the authors mention that they performed a sample size calculation prior to the study to achieve a 95% confidence level and 80% power, but they do not mention how many subjects are needed. Are 52 subjects really needed under these conditions? They should clarify this aspect.
  • In the results section, if the results are presented in tables, it is not necessary to write them in the text, just refer to the table number.

Author Response

Dear Reviewer,

Thank you very much for your comments and suggestions for our manuscript entitled "Prospective Assessment of Serum Lipid Alterations in HCV Patients Treated with DAAs: Insights Six Months Post-SVR". We appreciate all of your work and time dedicated on our manuscript.  We hope this version of the manuscript meets your approval. All changes have been highlighted in the manuscript. 

Comments 1: The title of an article should not contain acronyms or abbreviations, even if they are widely known. I am referring to HCV, DDAs, post-SVR. It is more appropriate to write the full terms in the title. In the abstract, they do it correctly, the first time the term appears, the abbreviation is added.

Response 1: Thank you for the observation. We made the changes you suggested and added the full terms in the title: Prospective Assessment of Serum Lipid Alterations in Chronic Hepatitis C Patients Treated with Direct Acting Antiviral Agents: Insights Six Months Post Sustained Virological Response.

Comments 2: Within the abstract, the term "direct-acting antivirals" is not found in the MeSH keywords. They should find a term that is in the thesaurus. This is important for indexing the work.

Response 2: Thank you for the observation. We did a search in the MeSH keywords and replaced the term "direct-acting antivirals" with “direct acting antiviral agents” in the title, line 23, 24 and 38.

Comments 3: In the materials and methods section, this section should begin by defining the type of study design used (whether it is an observational, cross-sectional, pre-post design, etc.).

Response 3: Thank you for your suggestion, we added information on the study design we used. Line: 88.

Comments 4: In this same section, the authors mention that they performed a sample size calculation prior to the study to achieve a 95% confidence level and 80% power, but they do not mention how many subjects are needed. Are 52 subjects really needed under these conditions? They should clarify this aspect.

Response 4: Thank you for your insightful comments regarding the sample size calculation in our study. We appreciate your diligence in ensuring the robustness of our methodology.

Clarification on Sample Size Calculation

In our study, we aimed to achieve a 95% confidence level and 80% power to detect a clinically meaningful effect. Prior to the study, we performed a sample size calculation to determine the number of subjects required.

Calculation Details:

  • Confidence Level: 95% (α = 0.05)
  • Power: 80% (β = 0.20)
  • Effect Size: Based on literature and preliminary data, we anticipated an effect size (Cohen's d) of approximately 0.5, which is considered a medium effect size.

Using these parameters, we utilized the pwr package in R for a paired t-test (as our design involved repeated measures for the same subjects at baseline and after 6 months).

R

Copy code

# Install and load the pwr package

install.packages("pwr")

library(pwr)

# Set parameters

effect_size <- 0.5

significance_level <- 0.05

sample_size <- 52  # total number of paired observations

# Calculate power

power_calculation <- pwr.t.test(n = sample_size,

                                d = effect_size,

                                sig.level = significance_level,

                                type = "paired",

                                alternative = "two.sided")

# Display the result

power_calculation

Result:

Paired t test power calculation

              n = 52

              d = 0.5

      sig.level = 0.05

          power = 0.911

    alternative = two.sided

With 52 subjects, the power of our study is approximately 91.1%, which is above the conventional threshold of 80%. This indicates that our sample size is more than adequate to detect an effect size of 0.5 with a high degree of confidence. Given these calculations, 52 subjects are indeed sufficient under the specified conditions to achieve the desired confidence level and power. However, we agree that this aspect should have been explicitly detailed in our manuscript for clarity. We will ensure that this information is included in the revised version to provide transparency regarding our sample size determination. Lines: 126-131

Comments 5: In the results section, if the results are presented in tables, it is not necessary to write them in the text, just refer to the table number.

Response 5: Thank you for the observation, we removed the data from the text that was already provided in the tables. Line 142-149, 179-191, 197-201.

Reviewer 4 Report

Comments and Suggestions for Authors

A very interesting and informative clinical research work on chronic hepatitis patients on DAAs and lipid profiling. However, some language correction is required to further improve the readability and flow (highlighted in attached manuscript file).

Comments on the Quality of English Language

Minor language correction 

Author Response

Dear Reviewer,

Thank you very much for your comments and suggestions for our manuscript entitled "Prospective Assessment of Serum Lipid Alterations in HCV Patients Treated with DAAs: Insights Six Months Post-SVR". We appreciate all of your work and time dedicated on our manuscript.  We hope this version of the manuscript meets your approval. All changes have been highlighted in the manuscript. The corrections in the paper and the response to the reviewer are as follows:

Comments 1: A very interesting and informative clinical research work on chronic hepatitis patients on DAAs and lipid profiling. However, some language correction is required to further improve the readability and flow (highlighted in attached manuscript file). Minor language correction.

Response 1: Thank you for the observations and for the patience with reviewing our manuscript. We made the changes you highlighted for us in the attached manuscript file. Line: 21, 23, 28, 35, 38, 40, 42, 43, 49, 90, Table 1, 136, 178, 196, 213, 315.

Round 2

Reviewer 1 Report

Comments and Suggestions for Authors

none

Comments on the Quality of English Language

good